# Venetoclax and Hypomethylating Agent Combination in Myeloid Malignancies: Mechanisms of Synergy and Challenges of Resistance

**DOI:** 10.3390/ijms25010484

**Published:** 2023-12-29

**Authors:** Rahul Mishra, Maedeh Zokaei Nikoo, Sindhusha Veeraballi, Abhay Singh

**Affiliations:** 1Department of Internal Medicine, Anne Arundel Medical Center, Annapolis, MD 21401, USA; rmishra@luminishealth.org; 2Department of Hematology and Oncology, Taussig Cancer Institute, Cleveland Clinic Foundation, Cleveland, OH 44195, USA; zokaeim@ccf.org (M.Z.N.); veerabs@ccf.org (S.V.)

**Keywords:** Venetoclax-5-Azacytidine, hypomethylating agents, acute myeloid leukemia, myeloid malignancies

## Abstract

There has been a widespread adoption of hypomethylating agents (HMA: 5-Azacytidine (5-Aza)/decitabine) and venetoclax (Ven) for the treatment of acute myeloid leukemia (AML); however, the mechanisms behind the combination’s synergy are poorly understood. Monotherapy often encounters resistance, leading to suboptimal outcomes; however, the combination of HMA and Ven has demonstrated substantial improvements in treatment responses. This study elucidates multiple synergistic pathways contributing to this enhanced therapeutic effect. Key mechanisms include HMA-mediated downregulation of anti-apoptotic proteins, notably MCL-1, and the priming of cells for Ven through the induction of genes encoding pro-apoptotic proteins such as Noxa. Moreover, Ven induces sensitization to HMA, induces overcoming resistance by inhibiting the DHODH enzyme, and disrupts antioxidant pathways (Nrf2) induced by HMA. The combination further disrupts oxidative phosphorylation in leukemia stem cells, amplifying the therapeutic impact. Remarkably, clinical studies have revealed a favorable response, particularly in patients harboring specific mutations, such as *IDH1/2*, *NPM1*, *CEBPA*, or *ASXL1*. This prompts future studies to explore the nuanced underpinnings of these synergistic mechanisms in AML patients with these molecular signatures.

## 1. Introduction

Acute myeloid leukemia (AML) is a heterogenous condition, generally affecting older individuals [1]. The ineligibility of such patients to intesive chemotherapy has led to the search for newer therapies in recent years. One such example of a newer combination of hypomethylating agents (HMAs) and a selective oral BCL-2 inhibitor (Venetoclax, Ven) has made significant strides and gained FDA approval in recent years [2].

HMAs, such as 5-azacytidine (Aza) and decitabine (Dec), function by promoting the blocking of DNA methyltransferase 1 (DNMT1). This process results in the demethylation of CpG islands [3] and an increase in the expression of genes necessary for normal myeloid hematopoiesis. HMAs, thus, play an integral role in the treatment of myelodysplastic neoplasms and acute myeloid leukemia (AML), when cellular differentiation blocks arrest maturation [4,5,6].

Normal cellular senescence via apoptosis, regulated by a balance between pro-apoptotic (Bax, Bak, Bad, Bid, Puma, Bim, Noxa) and anti-apoptotic proteins (Bcl-2, Bcl-xL, Bcl-w, Mcl-1), is lost among cancer cells, characterized by increased expression of anti-apoptotic protiens [7]. An anti-apoptotic protein prevents apoptosis by binding to and clearing pro-apoptotic proteins.

In 2014, Pan et al. [8] showed the efficacy of Ven (ABT-199), a potent, selective oral inhibitor of BCL-2, on multiple AML cell lines and patient-derived xenografts at nanomolar concentrations. Notably, the cytotoxic effects of Ven (Venetoclax) were predominantly unaffected by mutational status and were sustained in primary AML blast cells that were resistant to chemotherapy. In a phase II clinical trial, Konopleva et al. [9] showed evidence for the efficacy of Ven monotherapy (19% overall response rate), particularly among patients with *IDH1/2* mutations.

A phase 3 trial, the VIALE-A study, further explored the combination of these two agents in newly diagnosed older or unfit adults with AML. It showed high rates of composite complete remission in two-thirds of patients and superior survival and manageable toxicity with a combination of Ven + Aza, leading to the combination’s FDA approval in 2020 [2,10]. The combinatorial regimen has been successful in patients with relapsed or refractory (R/R) AML as well as among younger AML populations [11,12,13,14,15]. Understandably, there is widespread adoption and increasing usage of this highly effective regimen; however, the mechanisms behind the combination’s synergy are poorly understood.

In this review, we summarize the resistance mechanism to Ven or HMA monotherapy and highlight the putative synergistic mechanisms of the Ven–HMA combination in preclinical studies. We also review molecular signatures that predict clinical response to the Ven–HMA combination.

## 2. Mechanisms of Resistance against Monotherapies with Ven or HMA in AML Cell Lines

### 2.1. Resistance against Ven

The details of Ven resistance have been described elsewhere [16]. Briefly, the primary mechanism involves the overexpression of anti-apoptotic proteins other than BCL-2, i.e., BCL2-A1, MCL-1, and BCL-xL, that sequester pro-apoptotic proteins (e.g., BIM, BAX) [17,18,19]. Broadly, these mechanisms include the inability to trigger apoptosis due to inactivation of BAX/BIM, TP53, and PMA1P1 (NOXA); decreased BCL2 expression, hence lesser activity of Ven (a selective BCL-2 inhibitor); and alterations in mitochondrial function and cellular metabolism [20].

Another resistance mechanism includes stimulation of signaling pathways by mutant KRAS/PTPN11 or FMS-like tyrosine kinase 3 (FLT3) proteins. These mutations could either be intrinsic or emerge secondary to Ven-based therapy, and they carry an adverse prognosis [21]. The anti-apoptotic factors MCL-1 and BCL2-A1 are upregulated due to a mutation in the KRAS gene, whereas a mutation in PTPN11 results in the upregulation of the anti-apoptotic factors MCL-1 and BCL-xL. KRAS mutations also downregulate BCL-2 and BAX [19]. Venetoclax resistance is seen in FLT3-ITD mutations due to upregulation of the anti-apoptotic factors BCL-xL and MCL-1 via complex downstream pathways [22,23,24]. Additionally, preclinical data have provided evidence for Venetoclax resistance in AML with myelomonocytic differentiation (M4/M5 AML). This is hypothesized due to the relatively lower expression of BCL-2 and increased expression of MCL-1 and BCL2-A1 (through upstream mutant KRAS) in M4/M5 AML [8,19,20,21,22,23,24] (Table 1).

### 2.2. Resistance against HMA

The resistance mechanisms against HMA (Aza or Dec) are related factors intrinsic to tumor cells and extrinsic factors related to immune cells or other cells in the bone marrow, as discussed elsewhere [4]. Tumor cell-intrinsic factors are related to HMA transport (via human concentrative or equilibrative nucleoside transporter 1; hCNT1 or hENT1) into cells, activation (via kinases; UCK or DCK), incorporation into nucleic acids, and DNMT inhibition and metabolism via cytidine deaminases (CDAs). Low levels of transporters and decreased expression of kinases, leading to decreased incorporation of Aza or Dec into nucleic acids, contribute to HMA resistance [26,27,28,29].

HMAs can induce major antioxidant pathways mediated by the NF-E2-related factor (Nrf2), thus contributing to self-resistance [30]. Cheng JX et al. also reported that leukemia cells resistant to 5-Aza exhibit increased active chromatin organization associated with one of the methylcytosine (RNA: m^5^C) methytransferases, namely, NSUN1. NSUN1 establishes interactions with BRD4 and RNA polymerase II, resulting in the formation of an active chromatin structure resistant to 5-Aza but responsive to the inhibition of BRD4 [31]. Extrinsic factors are commonly related to, but not limited to, the bone marrow microenvironment. These are beyond the scope of this article and are discussed elsewhere [4].

## 3. Ven–HMA Synergy Mechanisms

### 3.1. Preclinical Data

#### 3.1.1. HMA-Mediated Downregulation of MCL-1 Levels

Tsao et al. [32] initially reported that co-administration of 5-Aza and ABT-737 (inhibitor of BCL-2 and Bcl-xL) could increase mitochondrial outer membrane permeability among AML cells. This was reflected via the activation of BAX protein and loss of the mitochondrial membrane potential. They noted that the 5-Aza and ABT-737 combination synergistically induced apoptosis independent of p53 expression in these cells, which was not the case when these agents were used as single agents. Elevated levels of anti-apoptotic MCL-1 are known for conferring resistance to ABT-737. The application of 5-AZA reduced the expression of MCL-1 in these cells, independent of p53, leading to enhanced AML cell apoptosis.

The role of MCL-1 inhibition (by S63845, a specific MCL-1 inhibitor) in synergizing venetoclax activity was further assessed by Hormi et al. [33]. They observed that despite relatively weaker expression of MCL-1 compared to BCL-2, S63845 prompted apoptosis in AML cells and exhibited a robust synergistic effect with Venetoclax. Notably, AML cells resistant to Venetoclax displayed heightened sensitivity to S63845, underscoring the potential role of MCL-1 upregulation as a resistance mechanism to Ven in AML [33].

#### 3.1.2. BCL2 Family Protein as 5-Aza-Sensitizing Targets

Bogenberger et al. [34] noted greater sensitization to 5-Aza (three- to fourfold) in AML cells with RNAi-mediated knockdown of BCL-xL. Furthermore, they focused on the involvement of three prominent anti-apoptotic proteins—BCL-2, MCL-1, and BCL-XL—in influencing cell proliferation and responsiveness to 5-Aza. Notably, the siRNA-mediated silencing of BCL-XL or MCL-1 substantially diminished viability across the majority of the examined AML cell lines. In contrast, the impact of BCL-2 siRNA was comparatively less pronounced. The silencing of BCL-xL strongly sensitized particular types of erythroid cells, while MCL-1 sensitized more broadly across AML cells.

Navitoclax (ABT-263, its tool compound was ABT-737), a combined inhibitor of BCL-xL, BCL-w, and BCL2 and Venetoclax (ABT-199, selective BCL2 inhibitor) were utilized to assess the relative potency of these agents in sensitizing the AML cells for 5-Aza. Higher concentrations of ABT-199 compared to ABT-737 were required for the enhancement of Aza activity. The association was dose-dependent with ABT-737 but not with ABT-199, except for MDS-L cells. A greater synergy was noted with ABT-737 than ABT-199 in most cell lines. The sensitization was mediated by enhancing apoptosis; hence, they concluded that ABT-737 had greater potency when compared to ABT-199 and in combination with 5-Aza. Furthermore, they performed BH3 profiling to correlate with clinical responses. The most effective BH3 metric was formed by considering values from both NOXA and BIM peptides. This combined metric proved useful in distinguishing clinical outcomes—differentiating between patients who responded positively (sensitive) and those who did not (resistant/refractory) [34].

#### 3.1.3. Combination Disrupts Energy Metabolism and Targets Leukemia Stem Cells (LSCs)

Extended exposure to azacitidine (oral; CC-486) harnesses sustained DNMT1 loss and sustained hypomethylation. This can lead to preferential targeting of immature LSCs, potentially addressing myeloid maturation arrests by facilitating differentiation [35]. Pollyea et al. [36] reported that the combination of Ven  +  5-Aza altered the tricarboxylic acid cycle, as indicated by a decrease in α-ketoglutarate levels and an increase in succinate levels in LSCs. This suggested a targeted inhibition of electron transport chain complex II, specifically in LSCs. In vitro modeling further validated the inhibition of complex II through the diminished glutathionylation of succinate dehydrogenase.

Together, these data suggest that the azacitidine-mediated differentiation of LSCs coupled with Ven-mediated programmed cell death further synergizes each agent’s activity. While the Ven–HMA combination targets oxidative phosphorylation in LSCs, there may be an additional role of HMA-mediated LSC differentiation that augments durable responses in AML patients.

#### 3.1.4. Reactive Oxygen Species (ROS)-Dependent Antileukemic Activity

Ven augments HMA by inhibiting Nrf2 antioxidant pathway activation, induced by HMA. This leads to oxidative killing of AML cells, via an indirect underlying mechanism. The NF-E2-related factor 2 (Nrf2) antioxidant pathway is the key cellular mechanism that prevents oxidative cell damage. In the presence of oxidative stress and the generation of ROS, the transcription factor Nrf2 dissociates from its cytoplasmic assembly with Kelch-like ECH-associated protein 1 (Keap-1). Under normal circumstances, Keap-1 facilitates the ubiquitination and degradation of Nrf2 [37,38]. Upon release, Nrf2 translocates to the nucleus, where it binds to the antioxidant response elements in the promoters of several genes, including heme oxygenase-1 (HO-1) and NADP-quinone oxidoreductase-1 (NQO-1). These genes play a role in neutralizing ROS [37,38].

Nguyen et al. [30] showed that in AML cell lines (MV 4–11), the combination of HMA (5-Aza and Dec) and Ven increased mitochondrial ROS production and apoptosis in comparison to HMA alone. They demonstrated that binding of Ven to Bcl–2 probably causes Bcl–2 to separate from its complex with Nrf2/Keap–1, which, in turn, increases the ubiquitination and Nrf2 degradation mediated by Keap–1.

Hu et al. [39] reported that the combination of a novel HMA (thio deoxycytidine, T-dCyd) with venetoclax kills myelodysplastic cells through an ROS-dependent mechanism. They noted increased ROS generation and downregulation of the antioxidant proteins Nrf2 and HO-1, as well as BCL-2. Furthermore, ROS scavengers, such as N-acetyl cysteine, reduced the lethality of the HMA–Venetoclax combination. A similar observation in AML cells is yet to be reported; however, considering the MDS to AML continuum, and clinically similar effects observed with HMA–Venetoclax in AML patients, such a contributory mechanism is possible.

Kamachi et al. [40] reported that the combination of novel oral HMA ORA2100 and Ven induced the downregulation of *VAMP7*, leading to an accumulation of ROS and inhibiting the growth of AML cell lines (HL60 and KG1a cells). VAMP7 is a member of the soluble N-ethylmaleimide-sensitive factor attachment protein receptor (SNARE) family and plays a role in the autophagic maintenance of mitochondrial homeostasis. The *VAMP7* gene knockout cells were noted to have attenuated growth and increased apoptosis relative to control cells. The *RNH1* gene plays a role in cellular redox homeostasis by catalyzing redox reactions. Both genes were significantly downregulated with the ORA2100-Ven combination. An increased accumulation of mitochondrial ROS was reported. Pre-treatment with a pan-caspase inhibitor, prior to application of ORA210 and Ven combination, led to a relative decrease in apoptosis (compared to without pre-treatment) but did not affect ROS accumulation. This indicated that ORA2100-Ven primarily triggered the accumulation of ROS, subsequently leading to apoptosis of AML cells and not vice versa. They also reported that Ven treatment mediated the increased expression of one of the anti-apoptotic proteins, MCL-1, in AML cell lines, but it did not contribute to OR2100-Ven therapy resistance. Interestingly, there was inverse correlation between baseline MCL-1 expression with sensitivity to the combination of OR2100-Ven.

#### 3.1.5. HMA (5-Aza) Induced “Priming” of the AML Cells for Ven-Induced Apoptosis

Jin et al. [41] demonstrated a novel non-epigenetic mechanism of action for 5-Aza and its synergistic activity with Ven through the integrated stress response (ISR)-mediated induction of *PMAIP1* transcripts. They reported that 5-Aza pretreated AML cells in vitro were more sensitive to Ven than DMSO pretreated cells. The number of cells with active caspase-3/-7 was higher immediately following treatment with combined 5-Aza and Ven exposure than with either agent alone. Molecular analysis revealed 5-Aza dose- and time- dependent increases in the pro-apoptotic protein factors NOXA (*PMAIP1*) and PUMA (*BBC3*) in cells obtained from AML patients. Yet, chronic 5-Aza therapy did not exert a substantial impact on gene expression or methylation. The promoter regions and transcripts of *PMAIP1* and *BBC3* genes remained unmethylated and unchanged. They postulated a role for the ISR pathway in Aza-induced priming. ATF4, the ISR pathway’s main effector, can promote apoptosis through transcriptional activation of proapoptotic factors (including *PMAIP1* (NOXA) and *BBC3* (PUMA)). When AML cells were exposed to the Ven–Aza combination, the levels of activating transcription factor 4 (ATF4) were significantly elevated within 24 h, indicating the activation of the ISR pathway by 5-Aza. These preclinical findings underscore the clinical importance of initiating both therapies concomitantly, preferably within 24 h of HMA initiation [41,42]. The relative impact of NOXA and PUMA was delineated by deleting their respective genes (PMAIP1 and BBC3). A reduction in the magnitude of cell death from the 5-Aza/Ven combination was observed in *PMAIP1*-deleted cells but not in BBC3-deleted cells. Therefore, they came to the conclusion that, whereas 5-Aza therapy induces both NOXA and PUMA, only NOXA is essential for sensitizing AML cells to 5-Aza/Ven-induced apoptosis [41].

Cojocari et al. further showed that the addition of Pevonedistat (Pev), a myeloid cell leukemia-1 protein (MCL1) inhibitor to Ven-5-Aza combination, augments NOXA induction to a greater degree than the addition of 5-Aza or Pev alone to Venetoclax therapy [43]. MCL-1 overexpression is one of the putative mechanisms of resistance to Venetoclax-based therapies in AML cells; hence, the Pev/5-Aza/Ven combination could be used as a novel therapeutic approach to overcome such Ven resistance. Notably, NOXA induction has been observed with decitabine in triple-negative breast cancer cells [44], enhancing the cytotoxic potential of cisplatin therapy in such cells.

#### 3.1.6. Overexpression or HMA-Mediated Restoration of Caspase-3/GSDME Significantly Increases Ven-Induced Pyroptosis

Ven induces pyroptosis through the cleavage of the gasdermin (GSDM) protein, a process mediated by caspase-3. Cleaved GSDM forms membrane pores, leading to cytokine release and/or programed lytic cell death, called pyroptosis [45]. Recently, Ye et al., via in vivo and in vitro experiments, showed that GSDME, but not GSDMA, GSDMB, GSDMC, or GSDMD, was cleaved upon Ven treatment. GSDME is suppressed in AML cell lines through promoter methylation, and reduced GSDME expression is strongly correlated with an unfavorable prognosis. They showed that the overexpression of GSDME, achieved through the demethylation of the GSDME gene or the restoration of GSDME expression mediated by HMA, markedly enhanced Ven-induced pyroptosis in AML [46].

#### 3.1.7. Ven Augments HMA via Inhibiting De Novo Pyrimidine Synthesis

Upregulated de novo pyrimidine synthesis is the major adaptive resistance mechanism against HMA (Aza and decitabine) in MDS/AML cells [47]. The mitochondrial membrane potential is required for the functioning of the mitochondrial enzyme DHODH, a key enzyme of the de novo pyrimidine synthesis in cells. Ven, by inhibiting BCL-2, depolarizes the mitochondrial membrane and, hence, inhibits the DHODH enzyme. At concentrations that do not induce apoptosis, both Ven and the DHODH inhibitor teriflunomide substantially reduce the levels of cytidine- and deoxycytidine triphosphate (CTP, dCTP) in AML cells. This contributes to the downregulation of de novo pyrimidine synthesis, hence overcoming resistance against hypomethylating agents (HMAs) [48,49] (Table 2).

**Table 2 ijms-25-00484-t002:** Synergy mechanisms reported in preclinical studies.

Azacytidine (5-Aza) Supports Venetoclax (Ven)	
Tsao et al. [32]	HMA mediated downregulation of MCL-1 5-Aza down-regulates MCL-1 in a p53-independent manner, leading to enhanced apoptosis in AML cells when combined with ABT-737 (inhibitor of BCL-2 and BCL-xL)
Jin et al. [41]	Priming of AML cells for Ven-induced apoptosis 5-Azacitidine induces “priming” through integrated stress response -mediated induction of *PMAIP1* (gene for NOXA protein) transcripts, sensitizing AML cells to Ven-induced apoptosis
Ye et al. [46]	HMA mediated restoration of Caspase-3/GSDME significantly increases Ven-induced pyroptosis Venetoclax triggers pyroptosis in AML cells by cleaving GSDME, which is downregulated due to promoter methylation in AML cells and is associated with poor prognosis. GSDME overexpression, achieved through gene demethylation or HMA treatment, enhances Venetoclax-induced pyroptosis in AML.
Ven supports 5-Aza	
Bogenberger et al. [34]	BCL-2 family proteins in 5-Aza sensitization RNAi-mediated knockdown of BCL-xL sensitizes AML cells to 5-Aza, with BCL-xL and MCL-1 playing a crucial role in sensitization. Navitoclax (combined inhibitor of BCL-2, Bcl-xL and Bcl-w) was reportedly more potent than Ven (a selective BCL2 inhibitor) in enhancing 5-Aza activity, highlighting the significance of targeting BCL-xL, BCL-w, and BCL2
Gu et al. [48]	Ven mediated inhibition of de-novo pyrimidine synthesis Upregulated de-novo pyrimidine synthesis, mediated by key mitochondrial enzyme DHODH, is the major adaptive resistance mechanism against HMA. Ven, by inhibiting BCL-2, depolarizes the mitochondrial membrane and hence inhibit the DHODH enzyme, leading to overcoming the resistance against HMA.
Novel mechanism with Ven + Aza combination	
Pollyea et al. [36]	Disruption of TCA cycle in leukemia stem cells Ven + 5-Aza disrupts the tricarboxylic acid cycle, targeting oxidative phosphorylation in leukemia stem cells
Nguyen et al. [30] Hu et al. [39] Kamachi et al. [40]	Reactive oxygen species-dependent mechanisms
	HMA (5-Aza and decitabine) induces antioxidant Nrf2-pathway. Binding of Ven to Bcl-2 likely leads to the dissociation of Bcl-2 from its complex with Nrf2/Keap-1, thereby enhancing Keap-1-mediated ubiquitination and degradation of Nrf2. Hence, Ven augments mitochondrial ROS induction and apoptosis compared with HMA alone. Combination of thio-deoxycytidine and Ven kills MDS cells through an ROS-dependent mechanism, involving increased ROS generation and downregulation of antioxidant proteins. The VAMP7 (maintains mitochondrial autophagic homeostasis) and RNH1 (maintains cellular redox reactions) genes were significantly downregulated with the ORA2100-ven combination. An increased accumulation of mitochondrial ROS was reported leading to inhibition of growth of AML cell lines

### 3.2. Clinical Data: Molecular Predictors for Response to Ven–HMA Combination

The FDA approved Ven–HMA as a frontline treatment for older adults aged 75 years or above, or those with comorbidities that make them unsuitable candidates for intensive induction chemotherapy [2,50]. In a pivotal randomized clinical trial, there was a significant improvement in the median OS (14.7 months for Ven + HMA vs. 9.6 months for placebo + HMA) and complete response rates (37% vs. 18%, respectively) [10]. Since then, several studies have attempted to delineate molecular factors predicting outcome with Ven–HMA, hence predicting patient subgroups who would benefit maximally from this therapy (Table 3).

The analysis of data from a phase 1b/2 study involving 209 patients treated with Ven 400 mg or 600 mg in combination with either HMA or LDAC revealed a CR rate of 83.7% for pts with *IDH1/IDH2* mutations, 84.6% for patients with *NPM1* mutations, 59.5% for patients with *TP53* mutations, and 53.3% for patients with *FLT3* mutations [51]. In the pivotal VIALE-A trial [10] (*n* = 431, median age of 76-years), patients with *IDH1* or *IDH2* mutations at baseline had better OS at 12 months, with 5-Aza + Ven (66.8%), as compared to 35.7% with 5-Aza + placebo group (with the hazard ratio for death, 0.35; 95% CI, 0.20 to 0.60; *p* < 0.001). Superior responses were noted with Ven + HMA in patients with *IDH1/2*, *FLT3*, *NPM1*, and *TP53* genes. In another study, DiNardo et al. reported high response rates and sustained remissions among patients with *NPM1* or *IDH2* mutations (2-year OS of 71.8% and 79.5%, respectively). The study identified that primary and adaptive resistance to Ven-based combinations was predominantly associated with mutations in *FLT3, RAS*, or *TP53* genes [52]. Further, a pooled analysis of data from clinical trials confirmed that patients with *IDH1/2* wild-type acute myeloid leukemia (AML) and subjected to treatment with Ven + 5-Aza exhibited less favorable results when presenting with poor-risk cytogenetics in comparison to their counterparts with *IDH1/2* mutations. Patients with *IDH1/2* mutations demonstrated superior outcomes, irrespective of cytogenetic risk, with a median overall survival (mOS) of 24.5 months in the intermediate-risk category, and mOS was not reached in the poor-risk category. Conversely, those with *IDH1/2* wild-type AML experienced a median overall survival of 19.2 months in the intermediate-risk group and 7.4 months in the poor-risk group [13].

**Table 3 ijms-25-00484-t003:** Genomic predictors for response to Venetoclax–Azacytidine combination reported in clinical studies.

Study	Type of Study	Population	Favorable Predictors	Unfavorable Predictors	Key Findings
Stahl et al. [11]	Retrospective	86 patients with relapsed/refractory AML	*NPM1* gene mutations	Adverse cytogenetics, *TP53, KRAS/NRAS, SF3B1* mutations	Higher response rates and OS with azacitidine + venetoclax. *NPM1* mutations associated with better response. Adverse cytogenetics and mutations in *TP53, KRAS/NRAS, SF3B1* associated with worse OS.
Morisa et al. [12]	Retrospective	86 patients (newly diagnosed and relapsed/refractory AML)	*CEPBA* mutation	-	*CEPBA* mutation favored CR/CRi. Higher overall survival for complete responders compared to non-responders.
DiNardo et al. [15]	Clinical trial	86 patients (44 newly diagnosed AML, 42 relapsed/refractory AML) frontline therapy in old patients with AML	*NPM1* or *IDH2* mutations	*FLT3* or *RAS* or *TP53* mutations	High response rates and durable remissions with NPM1 or IDH2 mutations. Primary and adaptive resistance characterized by *FLT3, RAS*, or *TP53* mutations.
Johnson et al. [53]	Retrospective	relapsed/refractory AML	*ASXL1* gene mutation	*TP53* mutations, absence of *IDH1/2* mutations, non-achievement of CR/CRi	*ASXL1* gene mutation predicted superior response. *TP53* mutations predicted inferior response.
Gangat et al. [54]	Retrospective	103 patients treatment naïve *AML*	*ASXL1* mutation, absence of *TP53* and *FLT3-ITD* mutations	Presence of *TP53* mutation	*ASXL1* mutation and absence of TP53 and *FLT3-ITD* predicted favorable response. *ASXL1* mutations and adverse karyotype predicted inferior survival.
Weng et al. [55]	Retrospective	150 Chinese patient population with relapsed/refractory AML	Mutations in *IDH1/2, NPM1, ASXL1,* chromatin–cohesin genes	Adverse cytogenetics, mutations in *FLT3-ITD, K/NRAS*	Adverse cytogenetics and ELN adverse risk predicted inferior response. Mutations in *IDH1/2, NPM1, ASXL1*, and chromatin–cohesin genes predicted superior response.

Morisa et al. retrospectively performed analysis of 86 patients (44 newly diagnosed with AML and 42 with relapsed/refractory AML) who received an HMA–Ven combination. In the upfront setting, logistic regression analyses revealed that the presence of the *CEPBA* mutation conferred improved CR/CRi rates. Each of the four patients with *CEPBA* mutation attained CR, in contrast to 51% (18 of 35) of *CEPBA* wild-type patients. The median OS for patients receiving upfront HMA–Ven was 11 months (95% CI; 8–23 months). It was higher among those recieving CR compared to those not achieving complete response (17 months vs. 3 months). For relapsed/refractory disease, the presence of *JAK2* (*p* = 0.03), *DNMT3A* (*p* < 0.01), and *BCOR* (*p* = 0.04) mutations was identified as predictors of CR/CRi. Median overall survival for relapsed/refractory AML patients was 5 months (95% CI, 3–9 months), which was higher among those patients achieving complete response (15 months vs. 3 months) vs. those not achieving CR/CRi [12].

In a retrospective study, Stahl et al. [11] analyzed outcomes with Venetoclax and HMA/LDAC therapy for 86 patients with r/r AML. 5-Aza + Ven resulted in higher response rates (49% vs. 15%; *p* = 0.008) and higher median OS (25 vs. 3.9 months; *p* = 0.003) compared with low-dose cytarabine + Ven. *NPM1* gene mutations were correlated with higher response rates, while adverse cytogenetics and mutations in *TP53*, *KRAS/NRAS,* and *SF3B1* were linked to worse OS. In a similar patient population, Johnson et al. [53] reported that detection of the *ASXL1* gene mutation and the absence of an adverse karyotype were indicative of a more favorable response. However, the presence of the *ASXL1* gene mutation did not influence overall survival. Instead, survival was adversely affected by the presence of TP53 mutations, the absence of *IDH1/2* mutations, and the failure to achieve CR/CRi.

Gangat et al., in another retrospective study involving frontline Ven + HMA therapy for patients with AML (*n* = 103), reported molecular predictive factors. A favourable response was predicted by the presence of the *ASXL1* mutation and absence of *TP53* and *FLT3-ITD* mutations. Among *ASXL1*-mutated patients, presence of the *TP53* mutation did not impact CR/CRi rates (100% vs. 81%). Conversely, among *ASXL1*-unmutated patients, the presence of the *TP53* mutation predicted poorer response (26% vs. 63%; *p* = 0.001). However, MRD positivity (60% vs. 13%) and relapse rates (73% vs. 35%) were higher in patients with *ASXL1*-mutated vs. *ASXL1*-unmutated status. The presence of *ASXL1* mutations and an adverse karyotype, coupled with the absence of CR/Ri, predicted poorer survivial outcomes [54].

A retrospective analysis of data for 150 Chinese patients with AML (with a median age of 53.5 years (range: 40.0–62.0), comprising 57 cases of primary refractory AML and 93 cases of relapsed AML) was conducted. Among the cohort, 57 patients had received prior chemotherapy, 36 patients were post-allogenic HSCT, and 25% patients had been previously exposed to HMA [55]. An unfavorable response to Ven–HMA combination therapy was predicted by adverse cytogenetics and the European Leukemia Net (ELN)-defined adverse risk category. Conversely, superior responses were associated with mutations in *IDH1/2, NPM1, ASXL1*, and chromatin–cohesin genes. The mutations in active signaling genes such as *FLT3- ITD and K/NRAS* predicted poorer response [55].

In a retrospective study of 35 patients with advanced MPN, MDS/MPN-overlap syndromes, or AML with extramedullary disease, Sanber et al. [56] reported outcomes with HMA/Ven therapy. The composite complete remission (CCR) was 42.9%, and the median OS was 9.7 months. A complex karyotype was linked to a significantly reduced median overall survival (3.7 vs. 12.2 months; *p* = 0.0002) and a lower CCR rate (22% vs. 50.0%; *p* = 0.244). Conversely, mutations in the SRSF2 gene were associated with an elevated CCR rate (80.0% vs. 28.0%; *p* = 0.0082) but did not exert an impact on median overall survival (10.9 vs. 8.0 months; *p* = 0.227) [56].

### 3.3. Superior Response with Ven–HMA Compared to Ven-LDAC

Clinical studies have reported superior response with Ven–HMA compared with Ven low-dose cytarabine (LDAC) therapy in newly diagnosed (CCR rate: 66.4% (Ven–HMA in VIAL-A trial) vs. 48% (LDAC-Ven in VIAL-C trial)) and r/r AML patients (CCR rate of 37% vs. 11%, respectively, in retrospective study) [10,11,57].

The precise mechanism of this difference in response rates is yet to be explored and likely extends beyond the different study populations in the two landmark studies that led to the FDA approval of both regimens. It relates to the mechanism of action or potential drug–drug interactions, patient characteristics, and specific genetic mutations, all of which can significantly influence response rates. HMA and Ven synergize together via multiple mechanisms (Figure 1). However, LDAC, also known as, arabinosylcytosine (ARA-C), undergoes conversion into its triphosphate form within the cell. This form competes with cytidine to integrate into the DNA. The sugar moiety of cytarabine impedes the rotation of the molecule within the DNA, leading to the cessation of DNA replication, particularly during the S phase of the cell cycle. Additionally, DNA replication and repair are halted due to the inhibition of DNA polymerase by cytarabine [58]. Variation in single nucleotide polymorphisms (SNPs) within the genes responsible for the transport, activation, and inactivation of cytarabine can impact the intracellular ara-CTP levels. These SNPs can influence the expression and activity of these genes and, consequently, can affect the clinical outcome of patients treated with ara-C [59].

Figure 1a shows that the intrinsic apoptotic pathway is regulated by proapoptotic (BAX, BAK, BID, BAD, BIM, PUMA) and anti-apoptotic (BCL-2, MCL-1, BCL-xL) factors. Pro-apoptotic activator proteins, once triggered by a stimulus, activate the central effectors of apoptosis (BAX and BAK) that undergo confirmational change and form pores in the mitochondrial membrane, causing miMOMP (minority-mitochondrial outer membrane permeabilization) and release of cytochrome-C. Free cytochrome-C then binds with Apaf-1 (apoptosome), which activates caspase-9 and triggers the apoptotic cascade. Anti-apoptotic proteins, when released, bind with and neutralize anti-apoptotic proteins. In normal cells, this balance is regulated via P53 proteins. However, in AML cells, there is predominant anti-apoptotic activity. Venetoclax, a BCL-2 inhibitor, promotes apoptosis in such cells. Hypomethylating agents (HMAs) synergize with Venetoclax (Ven) activities: (1) HMA (5-AZA) down-regulates MCL-1, a factor that can bind to Ven and attributed to Ven resistance. Thus, Ven is free to bind to the anti-apoptotic factor BCL-2, thereby preventing BCL-2-mediated downstream inhibition of pro-apoptotic pathways (Tsao et al., 2012 [32]). (2) HMA induces the pro-apoptotic factors NOXA and PUMA, although only NOXA is critical in sensitizing AML cells (Jin et al., 2020 [41]). (3) GSDME overexpression (by GSDME gene demethylation) or HMA-mediated restoration of Caspase-3/GSDME significantly increases Venetoclax-induced pyroptosis, another mechanism of programmed cell death (Ye et al., 2023 [46]).

Figure 1b shows that Venetoclax synergizes with HMA-mediated cancer cell killing: (1) The primary cellular mechanism for averting oxidative cell damage is the NF-E2-related factor 2 (Nrf2) antioxidant pathway. Exposure to oxidative stress and reactive oxygen species (ROS) leads to dissociation of the transcription factor Nrf2 from its cytoplasmic adaptor Kelch-like ECH-associated protein 1 (Keap-1). In the absence of oxidative stress, Keap-1 mediates the ubiquitination and degradation of Nrf2. Once released, Nrf2 translocate to the nucleus and activates genes, which mediate neutralization of ROS (Baird and Dinkova-Kostova, 2011 [38]; Nguyen et al., 2009 [37]). This pathway is also induced by HMA. Nguyen et al. (2019) [30] demonstrated that the binding of Venetoclax to Bcl-2 likely results in the dissociation of Bcl-2 from its complex with Nrf2/Keap-1. This, in turn, enhances the degradation of Nrf2, via Keap-1-mediated ubiquitination. Thus, the addition of Ven intensifies mitochondrial ROS induction and apoptosis, compared to HMA alone in AML cells. (2) The combination of novel oral HMA (ORA2100) and Venetoclax down-regulated VAMP7 (a member of the SNARE family, which plays a role in autophagic maintenance of mitochondrial homeostasis) and the RNH1 gene (plays a role in cellular redox homeostasis) and the accumulation of mtROS (mitochondrial ROS) (Kamachi et al., 2023 [40]). (3) Venetoclax can depolarize mitochondrial membranes. The membrane potential is required for the functioning of the mitochondrial enzyme DHODH that produces cytidine/deoxycytidine, which eventually compete with HMA in cells and are attributed to resistance against HMA. Thus, Ven prior to HMA dosing temporarily inhibits de novo pyrimidine synthesis and contributes to overcoming HMA resistance (Gu et al., 2020 [48]). (4) The anti-apoptotic protein inhibitors promote sensitization for HMA (three- to fourfold) by enhancing apoptosis. Higher sensitization for HMA was observed with ABT-737 (combined inhibitor of BCL-Xl, BCL-w, and BCL2) than with Venetoclax (selective BCL-2 inhibitor) (Bogenberger et al., 2014 [34]). (5) The Ven–HMA combination disrupted the tricarboxylic acid (TCA) cycle manifested by decreased α-ketoglutarate and increased succinate levels, suggesting the inhibition of electron transport chain complex II selectively in leukemia stem cells (Pollyea et al., 2018 [36]).

## 4. Conclusions

The combination of HMA–Ven has emerged as a promising and effective therapeutic strategy for older or unfit adults with myeloid malignancies. In this review, we highlighted the evolution of this combination (starting from the resistance mechanisms to Ven or HMA monotherapy) from preclinical studies to its FDA approval in 2020, emphasizing its success in both newly diagnosed and relapsed/refractory AML populations. The preclinical data presented in this review elucidate several synergistic mechanisms of the Ven–HMA combination. It is clear that the combination synergizes in more than one way. Notably, the down-regulation of MCL-1 by HMAs and the disruption of energy metabolism and targeting of leukemia stem cells by the Ven–HMA combination play an important role in promoting the observed synergy. The induction of ROS-dependent antileukemic activity and the “priming” of AML cells for Ven-induced apoptosis by HMAs further contribute to the combination’s efficacy. Ven and HMA augment each other in a subset of the population with specific molecular aberrations. Certain molecular mutations (*IDH*, *NPM1*, *CEBPA,* and *ASXL1* mutations) have been found to be predictors of good response to combination therapy in clinical studies. Numerous other factors, such as monocytic leukemia phenotype and signaling pathway mutations, confer resistance. Further exploration and understanding of these nuanced underpinnings of negative predictors in myeloid malignancies will help inform effective precision trials in the future. Similarly, the exact mechanisms contributing to synergy require further mechanistic studies to elucidate putative foundations for future clinical trials.

## Figures and Tables

**Figure 1 ijms-25-00484-f001:**
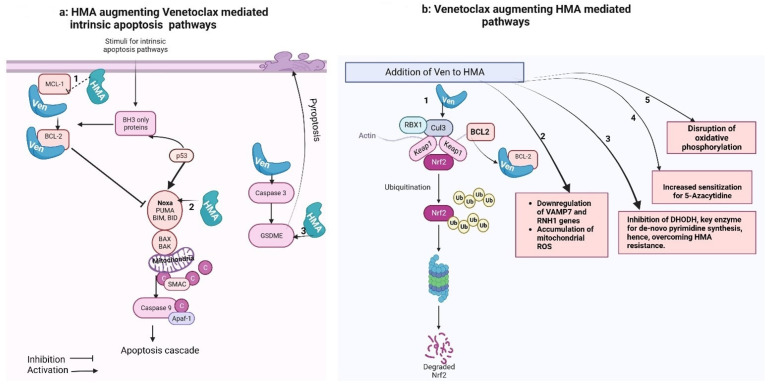
Venetoclax–hypomethylating synergy mechanisms. These figures were made using Biorender website (Biorender.com).

**Table 1 ijms-25-00484-t001:** Impact of common mutations on Venetoclax (Ven)–hypomethylating agent (HMA) combination response in AML patients.

Genetic Mutation	Response to Ven–HMA Combination	Proposed or Reported Mechanism
*NPM1*	Predicts positive response	Unclear mechanism.
*IDH1/IDH2*	Predicts positive response	IDH1/2 mutations induce neomorphic enzyme activity, producing R-2-hydroxyglutarate that inhibits cytochrome c oxidase, leading to apoptosis activation through BAX and BAK. However, the anti-apoptotic gene BCL-2 could antagonize BAX and BAK, thereby preventing apoptosis [25]. The BCL-2 antagonism by Ven counteracts this, promoting leukemic cell death and may explain the positive response in IDH1/2 mutant AML with Ven–HMA therapy. [13]
*KRAS*	Predicts resistance	KRAS mutation causes upregulation of MCL-1 and BCL2A1, downregulates BCL-2 and BAX. Contributes to Venetoclax resistance by upregulating anti-apoptotic proteins [19].
*PTPN11*	Predicts resistance	PTPN11 mutation causes upregulation of MCL-1 and BCL-xL. Contributes to Venetoclax resistance by upregulating anti-apoptotic proteins [19].
*FLT3*	Predicts resistance	FLT3-ITD mutation confers Venetoclax resistance by upregulation of BCL-xL and MCL-1 through complex downstream pathways [21,22,23].

## Data Availability

Not applicable.

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
