# Peer review of "Venetoclax and Hypomethylating Agent Combination in Myeloid Malignancies: Mechanisms of Synergy and Challenges of Resistance"

_ijms, 2023, doi:10.3390/ijms25010484_

Round 1

Reviewer 1 Report

Comments and Suggestions for Authors

Mishra and colleagues provide a very precise and accurate review of an aspect of great clinical importance, namely resistance to venetoclax and the mechanisms underlying the synergy between azacitidine and venetoclax. The review is well written and very enjoyable to read.

All aspects related to resistance to venetoclax known to date have been considered. Clinical data are precisely reported. The paragraph on the effect of venetoclax on the metabolism of leukemic stem cells is also well described. The bibliography is complete and updated.

Author Response

Thank you very much for taking the time to review this manuscript. We are sincerely grateful for your positive review and encouragement. 

Reviewer 2 Report

Comments and Suggestions for Authors

In their review  Mishra et al., analyze the mechanisms of synergy and resistance of venetoclax /5’Aza  co-treatment in AML patients.  Some sections are poorly argued and may seem difficult to follow.  Additionally, I believe because of  typo errors, in different sections the same factors is indicated as either  pro-apototic or anti-apototic, e.g. PMAIP1. However the review is interesting and highlights the intrinsic complexity and the need to stratify AML patients to maximize clinical benefit of the Ven/Aza treatment.

Reviewer 3 Report

Comments and Suggestions for Authors

In the manuscript „Venetoclax and Hypomethylating Agents Combination in Myeloid Malignancies: Mechanisms of Synergy and Challenges of Resistance” authors review preclinical and clinical data on the efficacy of combining venetoclax and HMA and further dissect their synergistic mechanism of action.

Broad comments: The manuscript is readable but there are some remarks to improve it

Specific comments:

1. What about low dose venetoclax and HMA or their combination? Are there any differences in the mechanism of action compared to regular doses? (eg. PMID: 37941406)

2. Table with clinical (trials) and preclinical data with experimental models and mechanism of action to sum it up would be useful.

3. Lines 128-139 – references missing.

4. Lines 153 and 156 – number of reference, further in the text aswell...

5. Are there any differentiative effects on leukemic blasts (except on LSC) of Ven+HMA combo?

Comments on the Quality of English Language

Moderate changes in language should be applied
